# A Scheme for Quickly Simulating Extraterrestrial Solar Radiation over Complex Terrain on a Large Spatial-Temporal Span—A Case Study over the Entirety of China

**Siwei Lin [1,2], Nan Chen [1,2,\*], Qianqian Zhou [1,2], Tinmin Lin [1,2] and Huange Li [1,2]**

[1] Key Laboratory of Spatial Data Mining & Information Sharing, Ministry of Education, Fuzhou University, Fuzhou 350116, China; n195520011@fzu.edu.cn (S.L.); 210310014@fzu.edu.cn (Q.Z.); lintingmin_66@163.com (T.L.); 132865@fzu.edu.cn (H.L.)

[2] The Academy of Digital China (Fujian), Fuzhou University, Fuzhou 350116, China

\* Correspondence: chennan@fzu.edu.cn

**Abstract:** Extraterrestrial solar radiation (ESR) is the essential basic background for solar radiation, which determines the occurrence of the weather and atmospheric phenomena. Since the influence of ESR variation on actual rugged terrain is a diverse, complex, and dynamic process, simulating ESR over a large spatial-temporal span, especially with a high-resolution digital elevation model (DEM), is a significant challenge. In this paper, we developed a new scheme for simulating ESR over the entirety of China using a DEM with a resolution of 30 m. To fully consider regional terrain status, the feature variables used were elevation, slope, and aspects of the located grid and the surrounding four grids to reveal the topography. In addition, latitude was used as a feature variable to consider the geographical location, and the month number was used to consider the duration. On the basis of different geographical locations, the training dataset was established from 20,000 grids. With the feature variable composition and training dataset, a backpropagation artificial neural network (BP ANN) was found to have the best performance compared with the other three machine learning methods in simulating ESR for a DEM. In terms of the proposed scheme and BP ANN, we drew an ESR map of China with a resolution of 30 m. The determination coefficient of the simulation result achieved 0.99 and the root-mean-square error was less than 50 MJ/m$^2$ in all sample areas, confirming its remarkable accuracy. In terms of efficiency, the time consumption of ESR simulated using the proposed scheme shrinks over 150 times in all sample areas compared to that simulated via the theoretical model. Simultaneously, the developed scheme was also used to simulate an ESR for a DEM with a resolution of 90 m to verify the universality and robustness of the developed scheme. In addition, we used the proposed scheme to derive the direct solar radiation and global solar radiation, thereby further proving the reliability and applicability of our study. Overall, our work convincingly proved that the proposed scheme is a potential and effective approach for quickly simulating ESR with high accuracy. This study provides the basis for different solar radiation inversions of long time series and large spatial scales, offering additional insights for simulating ESR on a large spatial-temporal span.

**Keywords:** extraterrestrial solar radiation; DEM; terrain shielding effect; spatial-temporal; machine learning algorithm

## 1. Introduction

Solar radiation, clean and cost-free energy, is the fundamental and dominating energy source in many different fields, and it can be converted into power (by photovoltaic power generation systems) [1], heat (by solar-thermal systems) [2], chemical energy (by various systems) [3], etc. Extraterrestrial solar radiation (ESR) is the basis for solar radiation [4,5], and refers to the maximum amount of solar radiation, considering the actual influence of terrain as its geographic counterpart without considering atmospheric attenuation [6]. It

determines the quantity and potential of energy resources on the surface receiving solar radiation [7], thereby further affecting the spatial distribution of other meteorological elements and surface fluxes.

ESR is the root of the Earth's atmosphere and the genesis of various weather phenomena [8]. As the decisive component for the ecosystem model of surface radiation equilibrium, ESR is a key parameter for deriving other types of solar radiation [9], including direct solar radiation (DSR) [10] and global solar radiation (GSR) [11]. Detailed data on ESR are required for the design of both photovoltaic and photothermal systems operating on Earth [12]. Moreover, it plays a vital role in a wide spectrum of fields, such as applications in agriculture, forestry, the building industry, and solar energy utilization [13–19]. The exploration of ESR, thus, is of increasing importance and has attracted considerable attention from researchers.

The digital elevation model (DEM) has become the most important source of practical space–Earth observation and it plays an irreplaceable role in scientific research [20], and it is a key data source for simulating ESR [21]. The DEM-based simulation of ESR from theoretical models is the main approach used for studying the regional spatial distribution of ESR, with high accuracy, in current research by scholars [22]. However, on the actual rugged terrain, different geographic conditions such as slope, aspect, and the latitude of the located site all exert great influence on the ESR quantity, especially the terrain shielding effects caused by the surrounding rugged terrains (see Figure 1). Though calculating extraterrestrial solar spectral irradiance at one moment is simple [23,24], the simulation of the sum of ESR quantity during a long period is complex and difficult due to the complex terrain influences [25].

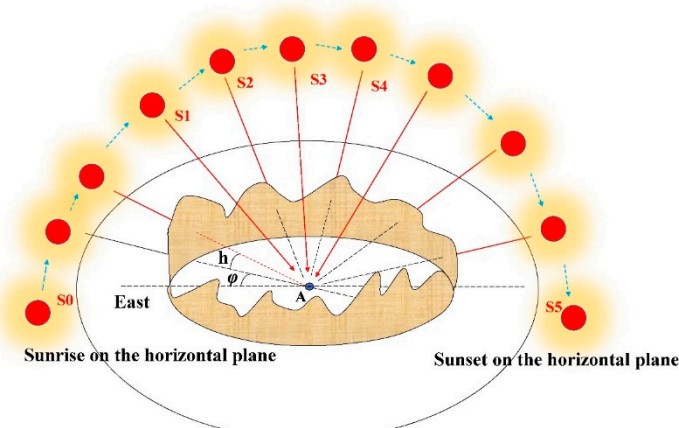

**Figure 1.** Illustration of terrain shielding effect on ESR. The process of sunrise to sunset on the horizontal plane is a process from S0 to S5. However, for a given position A, it may be shielded by the surrounding rugged terrain in different periods. The incidence direction was determined by solar azimuth angle $\varphi$ and solar elevation angle $h$. Take the process from S0 to S5 as an example, From S0 to S1, the incident sunbeam was blocked by the surrounding terrains that shielded A. At S1, A first receives sunbeams; at S2, shielding occurs; at S3, A receives sunbeams again; at S4, shielding occurs again. Clearly, the shielding status constantly varied with time, and the surrounding terrain environment exerts a strong influence on the ESR quantity of A in one period. In the meantime, $\varphi$ and $h$ in different periods also constantly vary, which makes the simulation of ESR in a period difficult to quantify using formulas.

For this, a distributed model, a powerful tool in the study of land surface processes [26], was proposed by scholars to consider the complex terrain influences on ESR in different periods [27]. In this case, it can be employed to simulate ESR over a long temporal span and has been widely used [27–32]. However, with the high accuracy brought by the distributed model, an inevitable issue that comes with it is the time-consuming and onerous work [14], especially on the large spatial scale or when using high-resolution data [31]. Due to

this issue, though massive numbers of scholars have systematically studied ESR spatial distribution in different regions, the simulation of ESR in current studies is limited to a small regional scale or to using low-resolution DEM data [28,31,33–35]. Thus, ESR simulations of large spatiotemporal distributions or high-resolution DEM data under current techniques are inefficient and time-consuming.

It is noted that solar radiation measurement has shown great importance in a variety of fields, such as climatology, meteorology, hydrology, pollution prediction, solar energy, agriculture, and material testing [25,36]. Nonetheless, we cannot ultimately derive high-resolution data of other types of solar radiation from ESR since ESR is difficult to simulate on a large spatial-temporal span or with high-resolution DEM data via present methods. Such technical limitations will cause difficulty in the decision making [37] and energy management [38] of the relevant fields mentioned above, which significantly constrains their development. In this case, how to efficiently and accurately simulate ESR in on large spatial-temporal span is a current bottleneck in this field.

In this paper, we develop a new scheme to simulate ESR over complex terrain on a large spatial-temporal span by considering the terrain information. We considered 17 feature variables for the ESR simulation of one grid, including the regional terrain (the slope, aspect, and elevation of the located grid and that of neighboring grids), geographic factors (located latitude), and time order (month number), and we conducted a DEM-based simulation of ESR using a machine learning frame. In terms of our proposed scheme, we tested and quantified the performance of the different machine learning models when simulating ESR on test sites, and found that the BP-ANN (backpropagation artificial neural network) performed best. Using BP-ANN, we simulated and delineated an ESR spatial distribution map of China to create a DEM with a 30 m resolution. Then, we discussed the performance of the proposed scheme and the ESR spatial-temporal distribution in China. Finally, we demonstrate the excellent function of the proposed scheme in deriving DSR and GSR data [39]. We have noted that the proposed scheme also performs well in a DEM with a resolution of 90m, which sufficiently confirmed its universality.

The rest of this paper is organized as follows. Section 2.1 introduces the study area and the distribution of sample areas. Sections 2.2 and 2.3 introduce the machine learning methods and experimental setup as well as the evaluation indices. In Section 3.1, we present the performance of the proposed scheme in different machine learning models and find the most appropriate machine learning framework. In Section 3.2, we simulate the ESR of China and fully describe it. In Sections 4.1 and 4.2, the simulated ESR is used to derive direct solar radiation (DSR) and global solar radiation (GSR) to validate the effectiveness of the proposed scheme. In Section 4.3, we used the proposed scheme to derive the ESR of China as part of a DEM with a resolution of 90 m. Section 4.4 comprehensively discusses the contribution of this study. Conclusions are drawn in Section 5.

## 2. Materials and Methods

### 2.1. Materials

In this study, we selected the entirety of China as our study area (see Figure 2). The DEM data used herein are the second version of Advanced Spaceborne Thermal Emission and Reflection Radiometer Global Digital Elevation Model (ASTER GDEM V2) with a resolution of 30 m, which was released to the public in January 2011. It can be downloaded from https://gdex.cr.usgs.gov/gdex/, accessed on 3 January 2022. ASTER GDEM data of Chinese land and corresponding islands were collected, with a total of 1120 images. Moreover, null values were filled using the nearest neighbor method. The relevant meteorological data of the 98 stations used in this study were provided by the CAM National Center of Meteorological Information, including the monthly mean sunshine percentage, monthly direct solar radiation, and global solar radiation quantity from 2006 to 2016.

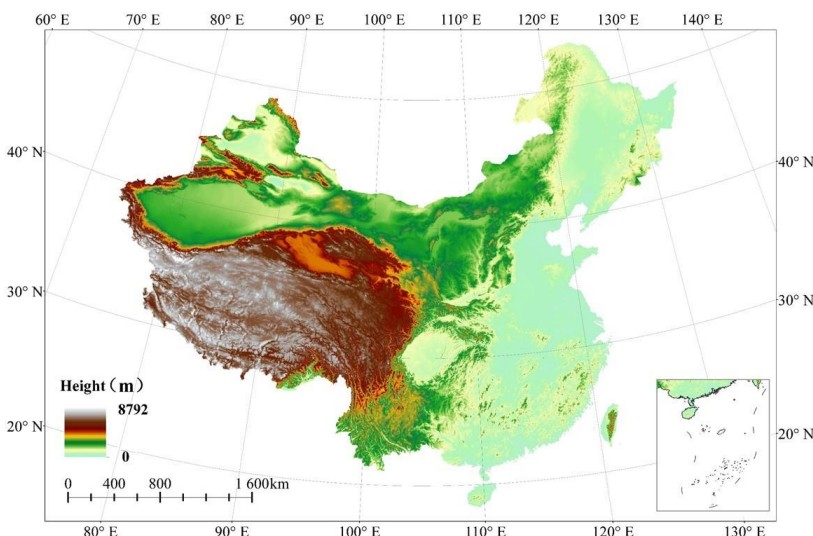

**Figure 2.** The study area of China.

*2.2. Different Machine Learning Algorithms Used in Simulating ESR*

A series of studies have demonstrated the good performance and great popularity of different machine learning methods in estimating, predicting, and extracting land or ocean information, including artificial neural networks (ANNs) [40–43], light gradient boosting machines (LightGBMs) [41,43–45], extreme gradient boosting (XGBoost) [46–48], support vector machines (SVMs) [49,50], etc. In this study, four machine learning methods were used to find an optimum model for predicting ESR by comparing their performances. The fundamental principle of these models is as follows:

(1)   BP ANN model

ANNs are one of the most highly used modeling techniques [51], and they are similar to the human brain, which is interconnected by different neurons [52]. Compared with the traditional statistical methods, ANNs exhibit greater abilities in simulating the nonlinear model [53]. A BP ANN using a backpropagation algorithm is a multilayer ANN structure that consists of an input layer, one or some hidden layers, and an output layer [54]. By using the backpropagation (BP) error for training, BP ANN can efficiently select the optimal weights and bias [55].

(2)   LightGBM model

LightGBMs are an improved method based on the gradient-boosted decision tree (GBDT) frame [56] (https://github.com/microsoft/LightGBM accessed on 3 January 2022). Two significant optimization techniques for improving the training speed with LightGBMs are the histogram-based algorithms and the leaf-wise tree growth strategy. When searching for the best segmentation point, a LightGBM uses histogram-based algorithms to transform traversal histograms instead of traversing samples, thereby greatly reducing the time and model complexity. Meanwhile, rather than the level-wise tree growth strategy used in selecting tree growth direction, LightGBMs adopted the leaf-wise strategy with a depth limit. This avoids unnecessary work when searching for leaves with fewer contributions, thus dramatically improving the training speed. Furthermore, the parallel learning support of LightGBMs also reduces the training time and memory footprint. On this basis, the improvement of LightGBMs brings higher efficiency and faster training speed than other methods [57,58].

(3)   XGBoost model

The extreme gradient boosting algorithm (XGBoost) is a cutting-edge ensemble learning algorithm known for its lower training time and high accuracy. On the basis of GBDT, XGBoost has made a series of optimizations in terms of the basic algorithm. XGBoost uses

the cumulative sum of the predicted values of a sample in each tree as the prediction of the sample in the XGBoost system [59]. A key improvement of XGBoost is that it introduces regular terms to the objective function, which effectively constrains the growth structure of the tree and simultaneously avoids overfitting problems [60]. In addition, XGBoost uses the second-order approximation of the loss function to speed up the descent of the loss function, thus accelerating the model iterations.

(4)    SVM model

SVMs are a typical generalized linear classifier with excellent generalization ability and global optimization [61]. SVMs aim to partition the feature space to obtain the maximum margin hyperplane [62]. With the kernel functions satisfying Mercer's condition, SVMs can map the low-dimensional input feature space to the high-dimensional output feature space, thereby accordingly turning a nonlinear regression into a linear regression [50,63].

*2.3. Experimental Setup and Evaluation Criterion*

Figure 3 shows the flowchart for simulating ESR using DEM data based on the machine learning method, which can be divided into three stages: dataset constructing, model training, and model selection, as well as ESR simulation.

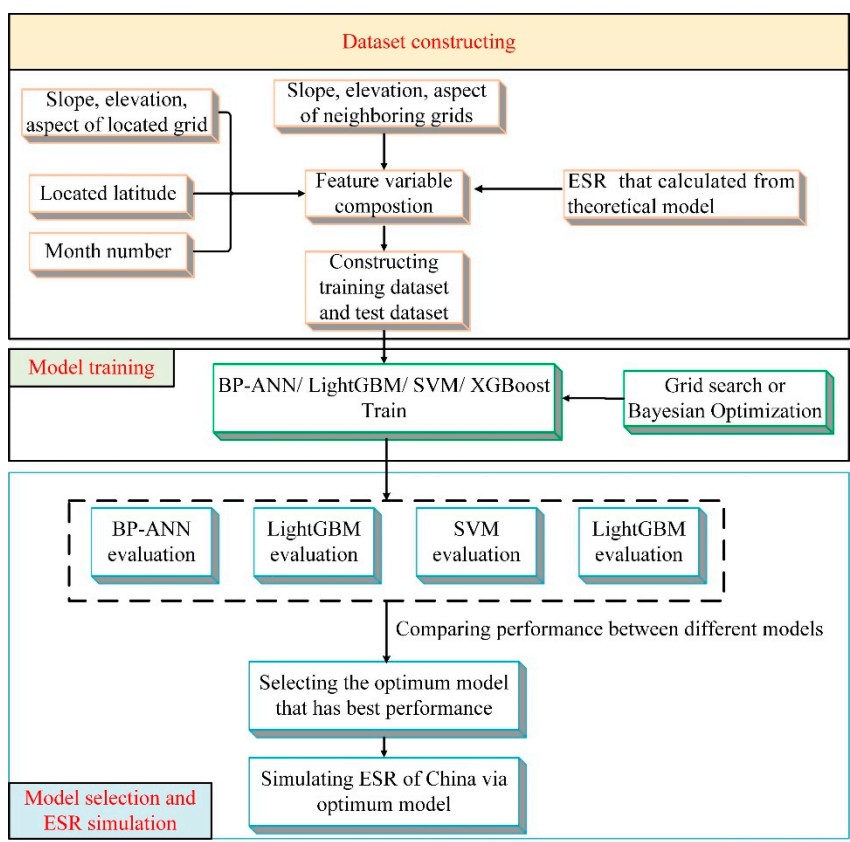

**Figure 3.** Flowchart for ESR simulation using DEM data based on the machine learning method.

2.3.1. Feature Variable Composition

The ESR quantity for a random site called A (see Figure 4) is an integrated result influenced by a series of space-time elements, including located terrain, surrounding terrains, geographical location, and duration. In our work, we comprehensively consider the above influencing factors of the ESR for each grid. Elevation, slope, and aspect, three typical terrain indices, are used as input variables to reveal the regional topography. Slope is a critical landform feature for quantitatively revealing terrain relief [64], which is closely associated with solar radiation distribution [32,65]. Aspect indicates the direction variation

of terrain, thereby strongly affecting the regional solar radiation quantity [29]. The slope and aspect for each grid are derived from a $3 \times 3$ matrix of neighboring elevations, which takes the grid as a center using the Horn algorithm (see Figure 5) [66,67]. In addition, considering the terrain shielding effect of surrounding terrains on ESR, the elevation, slope, and aspect of the four neighboring grids ($A_1$, $A_2$, $A_3$, and $A_4$) were also chosen as the input feature variables. Month number is adopted as a time parameter to determine the duration of ESR for A. Latitude was chosen as an input feature variable to determine the geographical position of A. We note that in previous studies for different types of solar radiation via DEM-based simulation, longitude is generally not a dominant factor [27,32,34,68,69]. For this reason, longitude is not considered in this article. To sum up, we herein considered 17 feature variables: elevation, slope, aspect for the located grid and the surrounding four grids (topography), month number (duration), and latitude (geographical location). In this paper, since we rapidly simulate the ESR on a monthly basis, the model output constructed using the BP-ANN is different for different months.

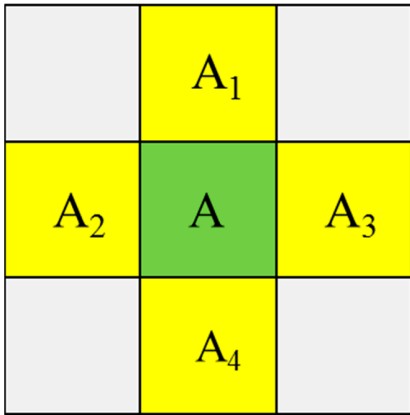

**Figure 4.** Grid A and its adjacent grids.

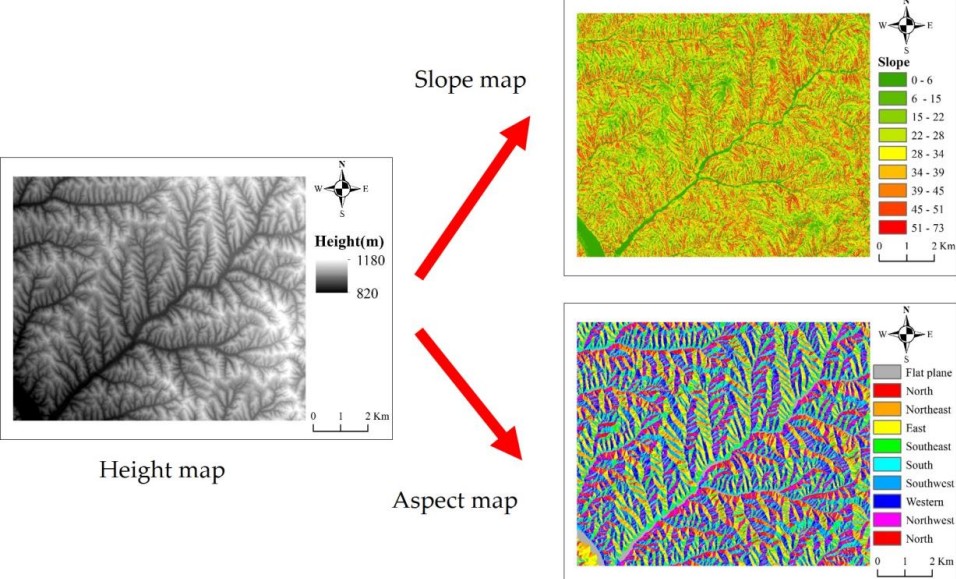

**Figure 5.** Basic dataset derived from DEM.

All the input feature variables were normalized within the boundary $[-1{-}1]$ for modeling [70]:

$$x_{i,j}^* = \frac{2\left(x_{i,j} - x_{imin}\right)}{x_{imax} - x_{imin}} - 1 \tag{1}$$

where $x^{*}_{i,j}$ is the normalized value; $x_{i,j}$ is the value before normalization; $x_{imax}$ is the maximum value of variable $j$; $x_{imin}$ is the minimum value of variable $j$.

### 2.3.2. Dataset Constructing

Given that the ESR is affected by the geographical location, 200 sample areas fully covering different latitude zones of China were selected. Then, ten grids randomly distributed among each sample area, totaling 20,000 grids, were selected as the basic dataset. The ESR value was obtained from the ESR theoretical model [8,27,31]. The dataset was divided into the training dataset and validation dataset by the ratio of 8:2.

Additionaly, to comprehensively estimate the performance of the training model, a test dataset is needed. Ten typical sample areas were selected from the 25°N, 30°N, 35°N, 40°N, and 45°N latitude zones in China. The sample areas covered the five basic landform types of China (plain, plateau, mountainous region, hill, and basin) in terms of the geomorphic division of Chen [71,72]. The sample areas were used as the test dataset for evaluating the model performance after training. Table 1 shows the information of the sample areas.

**Table 1.** Illustration for the ten sample areas.

| Number | Range of Latitude | Range of Longitude | Landform | Mean Slope (/°) |
|---|---|---|---|---|
| 1 | 25°N~26°N | 117°E~118°E | Low-middle mountain | 19.88 |
| 2 | 30°N~31°N | 88°E~89°E | High mountains of Himalayas | 16.03 |
| 3 | 30°N~31°N | 100°E~101°E | Hengduan Mountains, alpine and gorge region | 26.46 |
| 4 | 30°N~31°N | 105°E~106°E | Szechwan Basin | 10.95 |
| 5 | 35°N~36°N | 114°E~115°E | Northeast China Plain | 5.24 |
| 6 | 35°N~36°N | 108°E~109°E | Loess Plateau | 17.45 |
| 7 | 35°N~36°N | 117°E~118°E | Jerudong low hills and plains | 7.36 |
| 8 | 40°N~41°N | 81°E~82°E | Tarim Basin | 3.45 |
| 9 | 45°N~46°N | 119°E~120°E | Great Khingan middle and lower mountain | 8.10 |
| 10 | 45°N~46°N | 125°E~126°E | Songliao Plain | 6.50 |

### 2.3.3. Training Process

We trained different machine learning models with the training dataset and then validated the model and optimized the model parameters in the validation dataset. A grid search herein is used to find the optimum parameters of different machine learning methods [73,74] by fully traversing the combination of possible parameters. E.g., n_estimators (the number of residual trees), learning_rate, and num_leaves (controling the number of leaf nodes) are three key internal parameters for the LightGBM. Different scores were given by the combination of the three parameters under cross-validation. The optimal parameter combination was given by comparing the scores of different parameter combinations. In the BP-ANN model, Bayesian optimization was adopted to optimize the network parameters, which is an effective tool to avoid the over-fitting phenomenon in the training process and improve the generalization ability of the ANN model [75].

### 2.3.4. Model Selection and ESR Simulation

Finally, in the test dataset, we evaluate the performance of different models after training. The root-mean-square error (RMSE), mean absolute percentage error (MAPE), and correlation coefficient ($R^2$) were adopted to reveal the difference between predicted ESR and actual ESR (see Table 2). On the basis of these evaluation indexes, we can find the machine learning model with the best performance and use it to simulate the ESR of China.

**Table 2.** The adopted evaluation indexes.

| Model Evaluation Indexes | Formula |
|---|---|
| Mean absolute percentage error | $\text{MAPE} = \frac{1}{n} \sum_{i=1}^{n} \frac{|y_i - y_i^*|}{y_i} \times 100$ |
| Correlation coefficient | $R^2 = 1 - \frac{\sum_{i=1}^{n}(y_i - y_i^*)^2}{\sum_{i=1}^{n}(y_i - \bar{y}_i)^2}$ |
| Root-mean-square error | $\text{RMSE} = \sqrt{\frac{\sum_{i=1}^{n}\left(y_i - y_i^*\right)^2}{n}}$ |

## 3. Results

### 3.1. Performances of the Proposed Method Using Different Machine Learning Models

Figures 6 and 7 show the RMSE and $R^2$ of different models when simulating ESR in different sample areas. The BP-ANN, LightGBM, and XGBoost all showed good RMSE and $R^2$ scores in simulating ESR in the ten sample areas. The SVM showed poor simulation performance, with an RMSE far greater than that of the other three models, while the $R^2$ is the lowest. We note that the BP-ANN performed best in all sample areas, with an $R^2$ greater than 0.99 and an RMSE less than 50, which clearly confirms its outstanding performance. Thus, we selected the BP-ANN as the basic framework for the proposed method to simulate ESR.

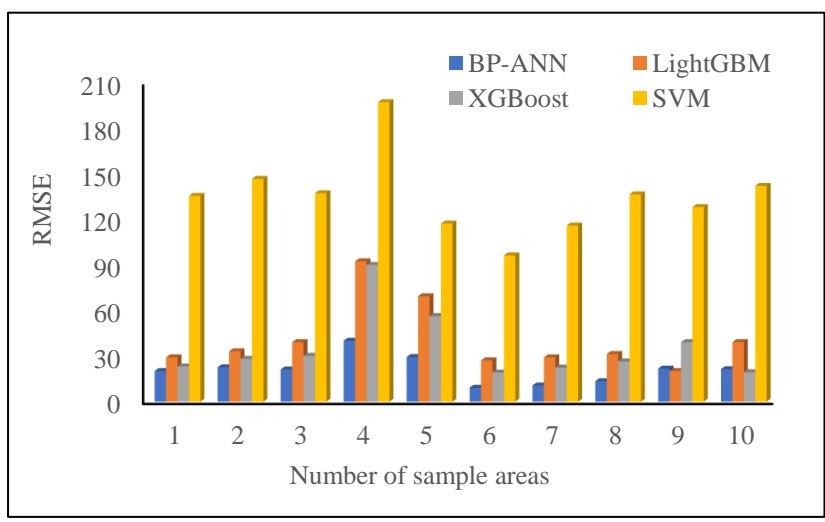

**Figure 6.** RMSE for different models in ten sample areas.

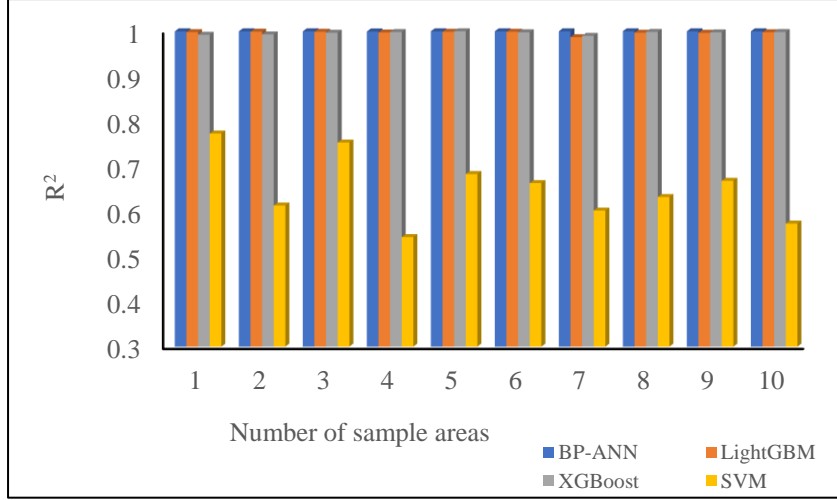

**Figure 7.** $R^2$ for different models in ten sample areas.

A Bayesian regularization algorithm is used to minimize the objective function and construct the BP neural network. The BP-ANN herein consists of an input layer, three hidden layers, and an output layer. The hidden layer structure is 13-13-13, in which the network error is smaller and the generalization ability of the model is better. Meanwhile, the transfer function of the hidden layer adopts the hyperbolic tangent S-type function tansig (), while the transfer function of the output layer adopts the linear function purelin (). The network structure is shown in Figure 8.

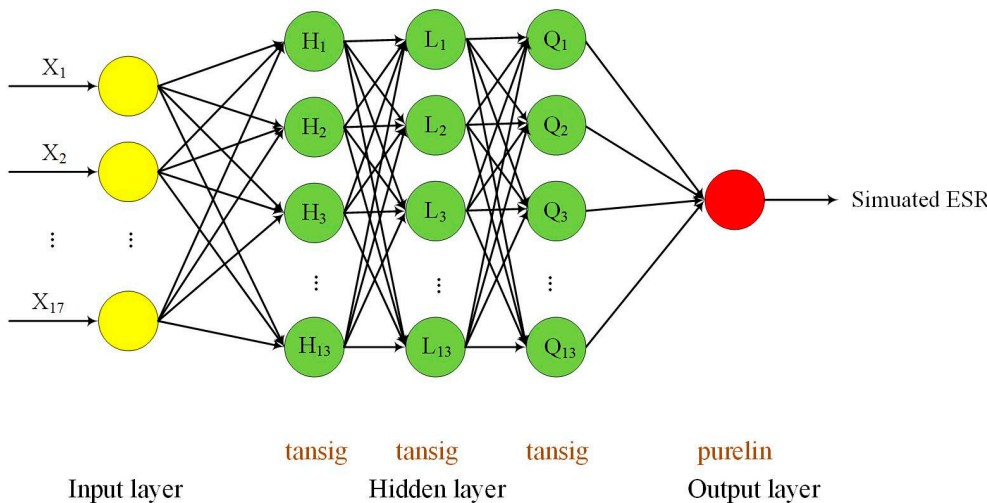

**Figure 8.** Optimum BP-ANN structure after Bayesian regularization.

So far, DEM-based studies on simulating ESR have generally adopted the theoretical model, which is a very time-consuming process. We compared the time consumption of the proposed method with the theoretical formula model for the ten sample areas (see Figure 9). The time taken by the theoretical formula model was 183~157 times larger than the proposed method, which demonstrates the extremely fast speed of the proposed method in simulating ESR. To further test the accuracy of the used BP-ANN, we measured the MAPE of ten sample areas in different months (see Figure 10). The MAPE is small from January to December and the simulation effect is stable in all months. That is, our method can efficiently simulate ESR while maintaining sufficient stability and accuracy.

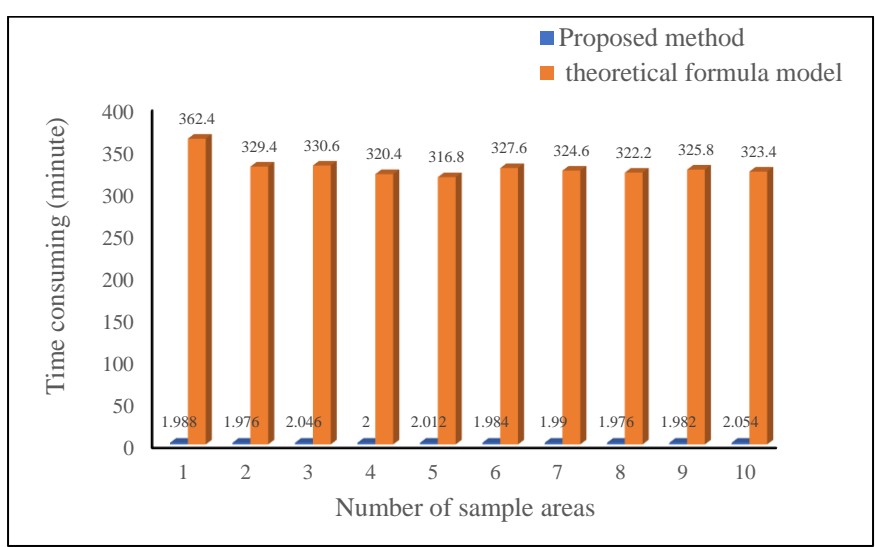

**Figure 9.** Comparison of the time taken to complete the simulation between our scheme and the theoretical model in ten sample areas. In this experiment, the computer had a CPU i7-6700, 64-bit operating system, and 16GB of ram as the basic configuration.

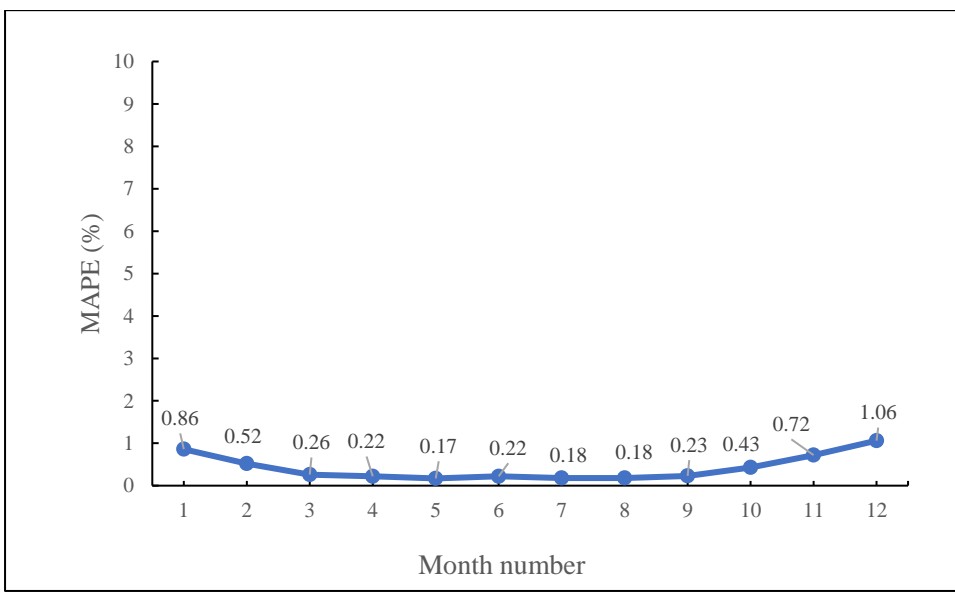

**Figure 10.** MAPE for BP-ANN in different months.

### 3.2. Simulating Spatial Distribution of ESR over China in Different Months

Using the proposed method, we simulated the spatial distribution of ESR in China from January to December (see Figure 11). Generally speaking, the spatial distribution of ESR is affected by temporal duration, topography, and latitude.

At the temporal scale, ESR showed a regular seasonal trend. ESR increased from spring to summer and reached a maximum mean value of 1216 MJ/m$^2$ in July, and then decreased from autumn to winter and reached a minimum value of 786 MJ/m$^2$ in December (see Figure 12). It is controlled by the movement of the subsolar point [76,77]. For example, the subsolar point in July just after the June solstice was in the closest neighboring region of the Tropic of Cancer, and the ESR quantity was highest for all of China; with the subsolar point moving southward, ESR gradually decreased. Correspondingly, the ESR quantity was the lowest in December, since the subsolar point was close to the Tropic of Capricorn. With the subsolar point moving northward, ESR gradually decreased. In total, the retention time rule of ESR quantity was consistent with the temporal distribution of heat resources in China [78].

On the other hand, we can note that the subsolar point also affected the spatial variation of the ESR spatial distribution, which is dominated by the latitude effect and terrain influence. As seen in Figure 13, the standard deviation (SD) [79] and variable coefficient (VC) [80] also showed a strong seasonal trend. The VC and SD reached their largest values in August and then decreased to the least value in January of the following year, which almost mirrors a process whereby the subsolar point moves from the summer solstice to the winter solstice. It may be explained as follows. As the subsolar point of the Sun moves southward, the solar altitude angle in the northern hemisphere becomes smaller, thus enhancing the terrain shielding effect on ESR. Alternatively, since China is in the northern hemisphere, the whole of China suffered from a weaker ESR, and the latitude differences of spatial distribution, therefore, become weaker. How the subsolar point of the Sun moves northward can be explained similarly.

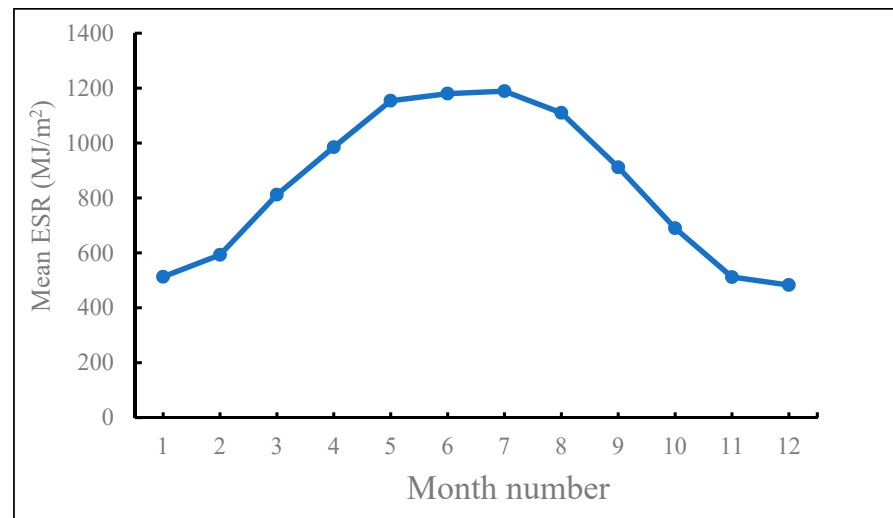

**Figure 11.** Spatial distribution of ESR in China in different months.

**Figure 12.** Mean value of ESR from January to December in China.

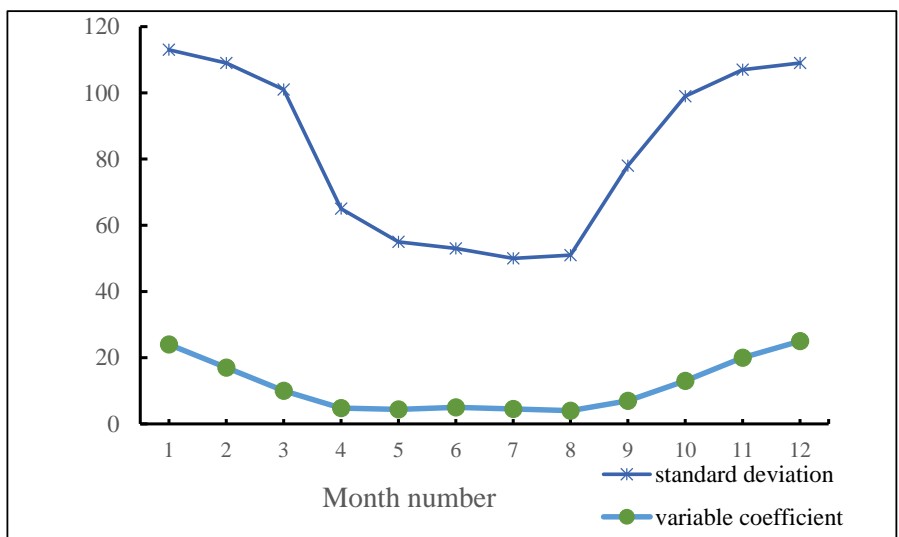

**Figure 13.** Variable coefficient and standard deviation of ESR in China from January to December.

At the spatial scale, ESR is comprehensively controlled by the located latitude and topography. We use the ratio of ESR quantity over complex terrain to the ESR quantity on the horizontal plane [30,81] to quantify the effect of topography on ESR, which can be given by:

$$R_b = \frac{Q_{0\alpha\beta}}{Q_0} \qquad (2)$$

where $R_b$ is the quantization factor, $Q_{0\alpha\beta}$ is the ESR over complex terrain, and $Q_0$ is the ESR on the horizontal plane. When $R_b < 1$, the ESR over rugged terrain is less than the ESR on the horizontal plane; when $R_b > 1$, the ESR over rugged terrain is greater than the ESR on the horizontal plane [27]. The closer $R_b$ is to 1, the weaker the influences of terrain on ESR are, and vice versa. The case that $R_b = 1$ indicates that the ESR quantity over complex terrain is equal to the ESR quantity on the horizontal plane, thereby further suggesting that ESR is not affected by terrain influences.

The terrain influences at different latitudes on ESR have different regularities. Figures 14 and 15 show the variation of $R_b$ with the aspect at different latitudes in January and July. The greater the latitude is, the greater the differences between $R_b$ and 1 in different aspects is. Namely, the higher the latitude of the sample area, the stronger the terrain's influence. In addition, we note that the amplitude of variation of $R_b$ in July (0.94~1.04) is relatively small, which indicated that latitude and terrain have little effect on ESR in autumn. On the contrary, the amplitude of variation for $R_b$ in January (0.5~1.9) is far greater, suggesting the great influences of latitude and terrain on ESR.

Alternatively, at the same latitude, the variation of $R_b$ under different slope and aspects is complex, embodying the complex influences of topography on the spatial distribution of ESR. In January, $R_b$ in the northeast and north is most affected by the slope, which gradually decreased with the increase in the slope (see Figure 16). In July, the $R_b$ values for different aspects were all less than 1 and had a decreasing trend with the increase in the slope.

We note that from an overall perspective of the whole of China, the terrain influence and latitude distinguishment for ESR is greatly obvious (see Figure 17). To conclude, two obvious characteristics can be found in the spatial pattern of monthly ESR. ① ESR shows latitude-dependent variation characteristics. That is, at the macro level, monthly ESR generally decreases as the latitude increased from south to north. Over different monthly durations, the variable extent is different, but it invariably existed at the macro level. ② ESR shows azonal distribution characteristics in mountains. In large-scale prominences with strong terrain relief, such as Hengduan Mountains, Tianshan Mountains, Kunlun Mountains, etc., the ESR distribution is generally significantly affected.

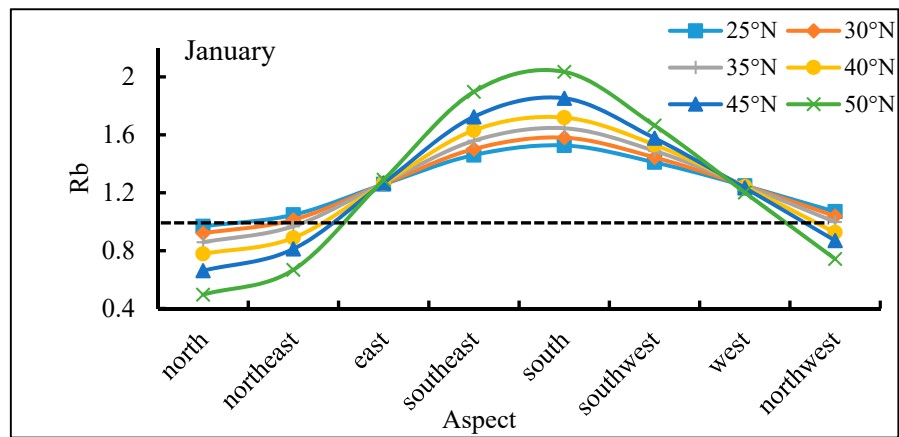

**Figure 14.** Variation of $R_b$ with aspect at different latitudes in January.

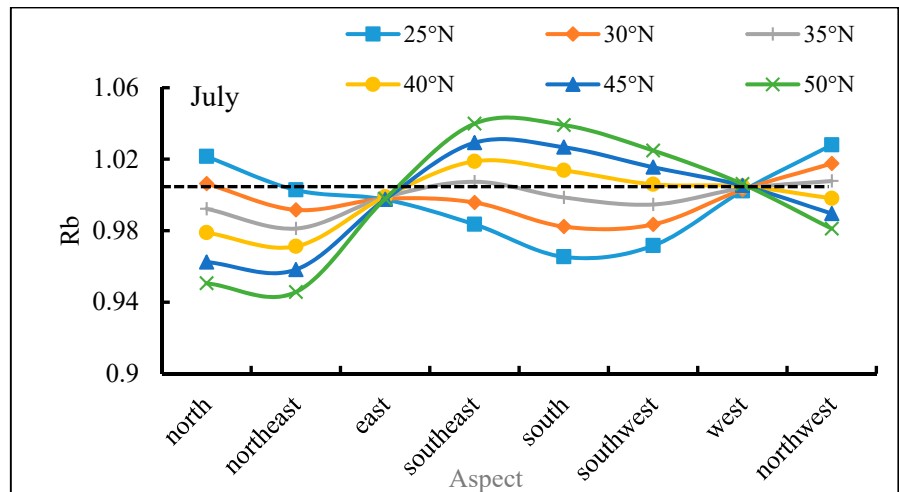

**Figure 15.** Variation of $R_b$ with aspect at different latitudes in July.

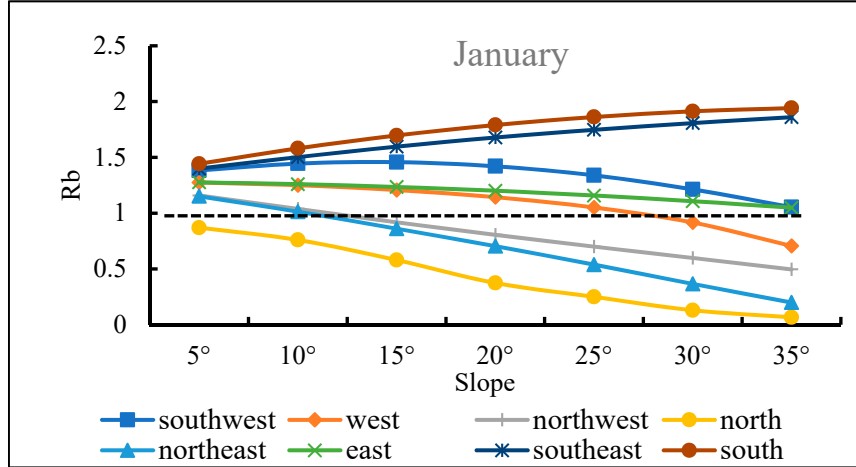

**Figure 16.** Variation of $R_b$ with terrain influence at a latitude of 30° in January.

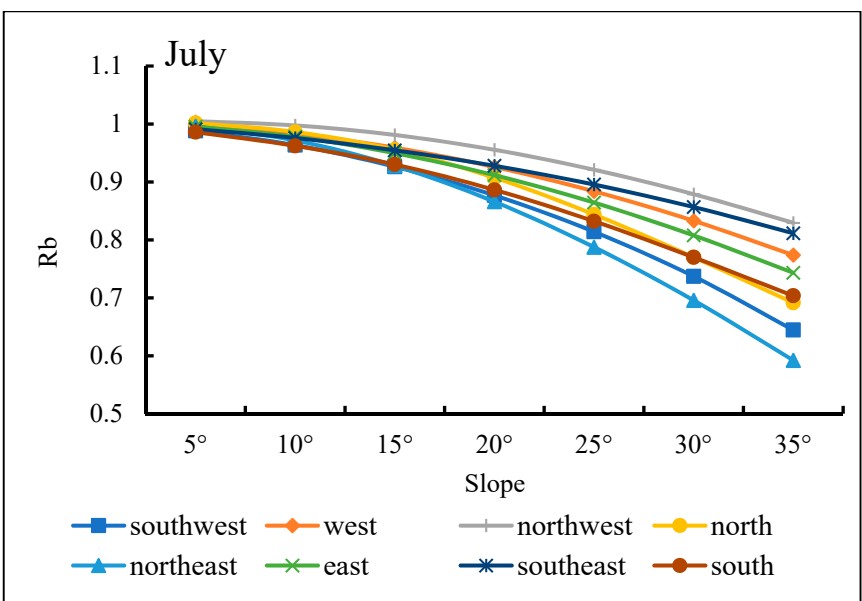

**Figure 17.** Variation of $R_b$ with terrain influence at a latitude of 30° in July.

## 4. Discussions

Measurements of surface radiation are too sparse to meet the demand for scientific research and engineering applications. It is impractical to obtain solar radiation information by setting up meteorological stations with wide coverage and near-real-time measurements on the Earth's surface. Simultaneously, monitoring techniques are generally time-consuming and costly, making them only applicable for monitoring on small spatial-temporal scales. To solve this problem, we herein investigated the performance of the proposed method in deriving other types of solar radiation information.

### 4.1. Proposed Scheme to Derive DSR

Direct solar radiation is closely related to agricultural production and is one of the main climatic factors that determines the ecological productivity for one region [10]. The experimental model for ESR is a mature method, which is widely used in simulating the regional DSR [10,68,69,82–84]. The simulation accuracy of the empirical model with ESR as initial data is better than that of other empirical models, which strongly shows its scientific significance [4].

Based on ESR data as the basic data source, the DSR can be quickly and simply derived. It can be given by:

$$Q_d = kQ_o = Q_o\left(as + bs^2\right) \tag{3}$$

where $k$ is direct transmissivity, $a$, $b$ are the empirical coefficients, $s$ is the sunshine percentage, and $Q_o$ is ESR given by the proposed method.

Therein, $s$ is given by meteorological observatories, and a, b are given according to the zoning model [5,82]. Table 3 shows the $R^2$ and MAPE of the derived ESR quantity in 98 meteorological stations. For the monthly DSR, the $R^2$ ranged from 0.96 to 0.97 and the MAPE ranged from 7.01% to 12.86%. For the annual DSR, the MAPE is 1.81% and the $R^2$ is 0.98, which convincingly proves the good performance of the proposed method in deriving the DSR.

**Table 3.** MAPE and $R^2$ for the derived DSR and measured DSR.

| Month Number | $R^2$ | MAPE/% |
|:---:|:---:|:---:|
| 1 | 0.97 | 9.35 |
| 2 | 0.96 | 10.11 |
| 3 | 0.96 | 12.86 |
| 4 | 0.97 | 9.67 |
| 5 | 0.97 | 9.96 |
| 6 | 0.97 | 7.01 |
| 7 | 0.97 | 8.76 |
| 8 | 0.97 | 11.31 |
| 9 | 0.97 | 8.46 |
| 10 | 0.97 | 12.11 |
| 11 | 0.97 | 7.32 |
| 12 | 0.97 | 9.35 |

### 4.2. Proposed Scheme to Derive GSR

Global solar radiation (GSR), the basis of weather and climate formation, is the underlying driver of the physical, chemical, and biological processes of the Earth's surface forces. ESR, as the basic data for combing the Angstrom formula [11], makes it possible to simulate the GSR with high precision on the real Earth's surface [85]. The experimental model using ESR to derive GSR is also an effective and fast method for simulating GSR, which has been extensively adopted by scholars [39,86,87]. Generally, it can be given by:

$$Q_g = Q_o(m + ns) \tag{4}$$

where $m$, $n$ are empirical coefficients from the zoning model [4], $s$ is the sunshine percentage, and $Q_o$ is the ESR that is given by the proposed method.

Table 4 shows the $R^2$ and MAPE of the derived GSR quantity in 98 meteorological stations. For the monthly GSR, the $R^2$ ranged from 0.90 to 0.96 and the MAPE ranged from 3.25% to 19.12%. For the annual GSR, the MAPE is 8.24% and the $R^2$ is 0.872, which shows reliable performance.

**Table 4.** MAPE and $R^2$ between the derived GSR and measured GSR.

| Month Number | $R^2$ | MAPE/% |
|:---:|:---:|:---:|
| 1 | 0.90 | 19.12 |
| 2 | 0.96 | 7.781 |
| 3 | 0.93 | 11.883 |
| 4 | 0.98 | 3.25 |
| 5 | 0.93 | 16.11 |
| 6 | 0.96 | 13.21 |
| 7 | 0.96 | 14.51 |
| 8 | 0.94 | 11.31 |
| 9 | 0.96 | 6.32 |
| 10 | 0.96 | 5.11 |
| 11 | 0.96 | 11.21 |
| 12 | 0.92 | 14.52 |

### 4.3. Deriving Different Resolutions of ESR in China

To validate the universality of our proposed scheme, the same scheme was also used for deriving the ESR on a DEM with the resolution of 90 m over China. The 90 m DEM data herein are from SRTM3 V4.1 [88], which can be downloaded from http://srtm.csi.cgiar. org/srtmdata/ accessed on 3 January 2022. With the same experiments on the same ten sample areas, Figure 18 shows the corresponding RMSE and $R^2$ for the ten sample areas. All the RMSE values were less than 40 and $R^2$ values were greater than 0.99, which proves

the feasibility of the proposed method in simulating ESR using the DEMs with different resolutions.

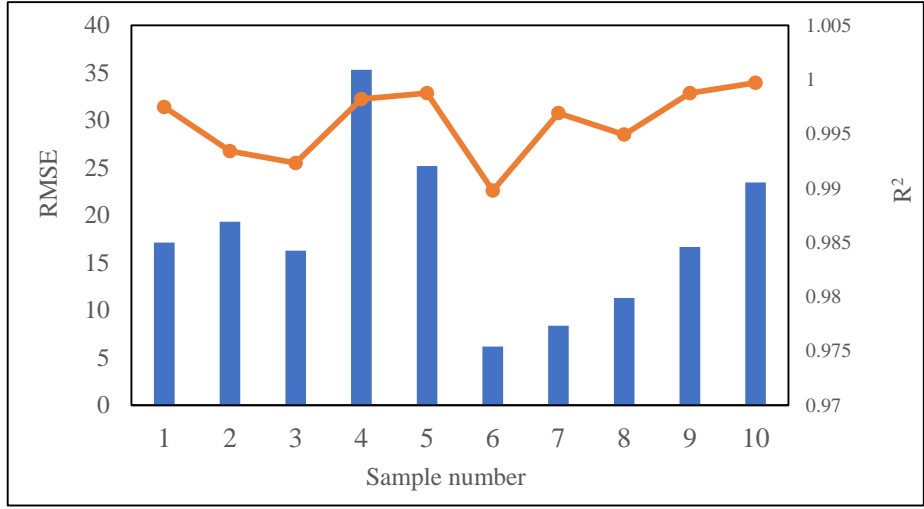

**Figure 18.** RMSE and $R^2$ in the same sample areas.

Figure 19 presents the comparison between the simulated annual ESR in DEMs with a 90 m resolution and a 30 resolution. Clearly, the spatial pattern of annual ESR in the DEMs of different resolutions was greatly influenced by the actual topography. Similarly, when based on a high-resolution DEM, the ESR data are more detailed. The DEM with a 30 m resolution can depict terrain in a more detailed and real way. In regions of strong terrain relief, the ESR quantity is lower, which is more in line with the actual surface condition.

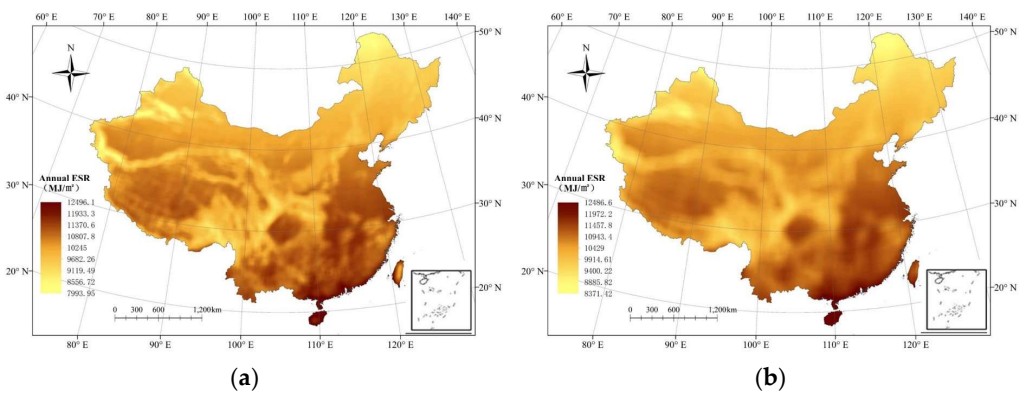

**Figure 19.** The comparison of ESR of China using DEMs with different resolutions. (**a**) Annual ESR with a resolution of 30 m. (**b**) Annual ESR with a resolution of 90 m.

### 4.4. Contributions of This Study

Though the popular distributed model has been extensively used in deriving ESR as a basic computational framework, simulating ESR on the large spatial-temporal span using it is hard. Since the distributed model comprehensively considers the terrain shielding effects caused by the actual rugged terrain, the price of high precision is heavy and time-consuming work. Especially in a high-resolution DEM, the calculation amount is huge and the corresponding heavy work may be beyond the capacity of researchers. On the other hand, with the development of new technologies, such as remote sensing and geographic information systems, the acquisition of DEM data with large spatial scales and high resolutions has gradually increased. In these cases, developing a method to quickly simulate ESR on a large spatial-temporal scale is imperative and necessary.

In this paper, we developed a method to simulate monthly ESR in the whole area of China. We developed a feature variable composition for simulating ESR and gave the

reasons for choosing them. Then, we constructed a training dataset that considers the geographical location. Finally, we tested the performance of different machine learning methods on simulating ESR and found that the BP-ANN performed best. The optimal model framework shows good simulation accuracy with high model efficiency, which significantly demonstrated its significant performance. In the meanwhile, by verifying the method on the DEM with a 90 m resolution, we further proved the universality and adaptability of the developed scheme. All of this work sufficiently suggests that the developed scheme provides new insights for solving the issue of simulating ESR on a large spatial-temporal span. In addition, by combing the mature experimental formula, the proposed method can effectively derive the DSR and GSR and show high accuracy, which fully extends the applications of the proposed method.

## 5. Conclusions

Due to the complex terrain shielding effect caused by the actual terrain relief, the calculation of the ESR quantity over a large spatial-temporal scale is challenging and inefficient. In this paper, we developed a method for combing the regional terrain indices to simulate the ESR on a large spatial-temporal span (the whole of China). We draw the following conclusion:

(1) The proposed scheme showed convincing performance in simulating ESR, including for different positions, landforms, and durations. The proposed method for combing with a BP-ANN is more advantageous in modeling the ESR quantity, with this model possessing the highest simulation accuracy, given that its $R^2$ values are all greater than 0.99 and its RMSE values are all less than 50. Simultaneously, compared with the previous method, the time consumption is reduced by nearly 200 times.

(2) The comprehensive spatial distribution of monthly ESR is caused by latitude, topography, and duration. The monthly ESR shows clear latitude-dependent variation characteristics and azonal distribution characteristics.

(3) The ESR from the developed scheme can be applied to rapidly derive DSR and GSR.

(4) The developed scheme is suitable for different resolutions of DEMs and showed good performances.

To summarize, a series of carefully designed experiments fully demonstrated that the developed scheme is a simple but effective method for simulating ESR on a large temporal-spatial span with excellent performance and fast speed. Its successful employment in deriving DSR and GSR and simulating ESR on the different resolutions of DEMs further deeply extends the application of the developed scheme, which may provide scientific guidance and important basic data for the simulation of DSR and GSR.

**Author Contributions:** Conceptualization, S.L. and N.C.; methodology, S.L.; software, S.L. and H.L.; validation, S.L.; formal analysis, S.L., N.C. and Q.Z.; investigation, Q.Z.; data curation, S.L.; writing— original draft preparation, S.L.; writing—review and editing, S.L.; visualization, T.L.; supervision, T.L. and H.L.; project administration, S.L.; funding acquisition, N.C. All authors have read and agreed to the published version of the manuscript.

**Funding:** This research was funded by the National Natural Science Foundation of China (grant numbers 41771423, 41491339, 41930102, and 41601408).

**Institutional Review Board Statement:** Not applicable.

**Informed Consent Statement:** Not applicable.

**Data Availability Statement:** The data that support the findings of this study are available from the corresponding author, Nan Chen, upon reasonable request.

**Conflicts of Interest:** The authors declare no conflict of interest.

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
