# Peer review of "A Scheme for Quickly Simulating Extraterrestrial Solar Radiation over Complex Terrain on a Large Spatial-Temporal Span—A Case Study over the Entirety of China"

_remotesensing, doi:10.3390/rs14071753_

Round 1

Reviewer 1 Report

The revised version of the manuscript addressed all my previous recommendations. No addition observations. 

Author Response

Thank you for your approval. Your kind consideration will be greatly appreciated

Reviewer 2 Report

The authors modified the manuscript as suggested by the reviewer. It seems that the article has gained a lot and is suitable for publication

Author Response

A lot of thanks for your approval. Your kind consideration will be greatly appreciated

Reviewer 3 Report

Dear Authors,

Thank you for your reconsideration of the paper. I understand that the number of explanatory variables increases, the ability of the model to explain the phenomena increases as well. However, the effects of the variables you used for the model (i.e., DEM, slope, aspect...) are not clear by means of their structures since those variables do not change within short time (e.g. one month, one year, a decade). These models are, eventually, fitting a line to a data cloud. Could you please provide a better statement to explain how precisely the model output can be used for the future researches by means of its suggested prediction algorithm.

Author Response

Comment

Thank you for your reconsideration of the paper. I understand that the number of explanatory variables increases, the ability of the model to explain the phenomena increases as well. However, the effects of the variables you used for the model (i.e., DEM, slope, aspect...) are not clear by means of their structures since those variables do not change within short time (e.g. one month, one year, a decade). These models are, eventually, fitting a line to a data cloud. Could you please provide a better statement to explain how precisely the model output can be used for the future researches by means of its suggested prediction algorithm.

Reply:

A lot of thanks for your careful consideration of the paper. Since the extraterrestrial solar radiation (ESR) is the maximum solar radiation considering the actual terrain influence in its geographic counterpart without considering atmospheric attenuation, ESR will not influenced by the meteorological environment. In this paper, since we rapidly simulate the ESR on a monthly basis, the model output constructed by the BP-ANN is different in different months. We added the explanation in lines 206 to 208. Please check. However, If you are referring to long time series predictions, then Long short-term memory neural network may be appropriate. 

This manuscript is a resubmission of an earlier submission. The following is a list of the peer review reports and author responses from that submission.

Round 1

Reviewer 1 Report

Dear Authors,

Please explain "-277 m." in figure 2.

Reviewer 2 Report

1. The article is written in a very hermetic language, so its scope (science impact) seems to be limited. Authors should make every effort to ensure that the manuscript is read by a broad scientific community. The reviewer recommends simplifying the language and adding a background for the research problem.
2. The summary includes the purpose and general conclusions of the research. The research method is missing.
3. In scientific research, the most important feature is research replication. The tests and the method should be described in such a way as to enable the repeatability of the tests and their results. The authors describe the newly presented method quite enigmatically. Perhaps a graphical diagram of the proposed method should be shown?
4. The authors already in the abstract suggest wide use of their original method. This should be explained and described.
5. Page 2, lines: 67-68 "it significantly constrains the development of related fields in practice." - what does it mean? This needs to be developed.
6. Abstract: "... RMSE was less than 50 in all sample areas." - 50 what? What are these units? "... performs the previous method, which shrinks the time-consuming ..." - what previous methods? Authors? Other researchers? Such very general information should not be included in the summary.
7. Section 2.3. Experimental Setup and Evaluation Criterion: "To sum up, we herein considered 16 feature variables: ...." - 16 variables accepted for analysis and in Figure 7 we have 17 entries to the network. Why? If bias is included, it must be clearly marked on the figure.
8. Section 3. Results - the methodology used to teach BP-ANN is very poorly described. What was the method of minimizing the objective function (eg Levenberg-Marquardt, conjugate gradients ...)? How was the data divided into the training set and the test set? The given results are for what harvest? Was the model validated during network training?
9. Section 3. Results: "In this case, we determined the superiority of our proposed method." - superiority over what? By other methods? So far used? Proposed by the Authors?
10. Overall, the methodology and conclusions differ from Abstract and Introduction. In the Introduction, the authors suggest that they will use different approaches in the field of artificial intelligence and create a new solution: "... complex terrain on a large spatial-temporal span via considering the terrain information. Via the comprehensive comparison between artificial neural network (ANN), Light Gradient Boosting Machine LightGBM (LightGBM), Extreme Gradient Boosting (XGBoost), support vector machine (SVM), we obtained the optimum framework for the proposed method. ”. And they describe these methods in section 2. Materials and Methods, and ultimately only use BP-ANN.
11. The authors write that they propose a scheme? But which? Neural networks with Bayesian regulation were used. Or maybe something more? Unfortunately, the authors do not write this.

Reviewer 3 Report

The paper deals with extraterrestrial solar radiation estimation problem by considering the terrain complexity and large spatial-temporal span, demonstrated for the case study of China. The authors propose a scheme based on BP ANN model able to accomplish the proposed target, i.e. estimation of direct and global solar radiation; the obtained results seem reliable and demonstrated high accuracy and speed of the proposed approach. The paper uses accurate English, in a clear and logical way.

The following issues are recommended to improve the paper:

  1. Define all the acronyms at their first use in Abstract and paper body, even they are well known in specific literature, e.g., DEM (in Abstract and Keywords), BP ANN, DSR,
  2. Solve several typing mistakes, e.g., “”area(see Figure 2)”; “dataset is needed. ten typical sample areas”; “As seen in figure 12”; “Figure 13. variation”; “Figure 14. variation”; “more different from 1 with the increase of slope”; “areas, figure 17 shows”; double spaces, etc.
  3. Abstract: “developed scheme combining the optimum machine learning method”. Introduce briefly the principles and novelty of the proposed scheme/method. Typically, “combining” is used for mixing at least two components (here is mentioned only a “method”)!
  4. “Solar radiation, a clean, cost-free, and endless energy, has emerged as a fundamental energy source in agricultural production and setting of photovoltaic power generation systems on earth [1].” Just few comments: theoretically, the solar energy is at universe scale a finite resource, only at humanity scale can be considered as “endless energy”. The solar energy can be converted into power (by PV systems), heat (by solar-thermal systems) and chemical energy (by various systems), not only for “setting of photovoltaic power generation systems”. The solar radiation is a fundamental energy source in various areas, not only in agriculture.
  5. Figure 1: depending on location latitude and day, generally the sunrise occurs in the Est direction proximity, not in the North direction! Keep the figure and its caption on the same page. “.. is a process from S0 to S6”, but S6 is missing on the figure.
  6. “data(R. Ambreen et al.,2011).” Use [12] to cite this work!
  7. End of Introduction: the novelty of the proposed approach should be briefly stated in an explicit way, in relation to the actual knowledge. Typically, the paper structure (Sections) should be briefly presented at the end of Introduction.
  8. “To sum up, we herein considered 16 feature variables: elevation, slope, aspect for the located grid and the surrounding four grids (topography), month number (duration), latitude (geographical location).” It seems there are 17 parameters, as 3 variables*5 grids + 2 = 17! See also Figure 7! Please also state/explain in the paper why the location longitude is not considered in the study!
  9. According to Eq. (1), the range of the normalize values is [-1,1], not (0,1)! Please, clarify it!
  10. (1): “????? is the maximum value of variable ?; ????? is the minimum value of variable j”, but the Eq. (1) contains only ?i?ax and ?i???. Why the indices i,j is used here in a general case of a variable value x???
  11. “movement of the subsolar point.” Define the subsolar point or cite a reference work!
  12. “standard deviation (SD) and variable coefficient (VC)”. Define SD and VC or cite a reference work!
  13. (2): explain the physical meaning of Rb < 1 and Rb > 1, respectively. As the complex terrain has a negative (decreasing) effect on ESR, why Rb can be over 1?
  14. Figure 14: use Rb on y axis title instead of RB.
  15. Use consequently North or north, South or south in body text and figures (Fig. 13, 14, etc.).
  16. Figure 18: translate (or eliminate) the text in Chinese.